# Axiomatic Explanations for Visual Search, Retrieval, and Similarity Learning

**Mark Hamilton**[1,2]**, Scott Lundberg**[2]**, Stephanie Fu**[1]**, Lei Zhang**[2]**, William T. Freeman**[1,3]
[1]MIT, [2]Microsoft, [3]Google
`markth@mit.edu`

## Abstract

Visual search, recommendation, and contrastive similarity learning power technologies that impact billions of users worldwide. Modern model architectures can be complex and difficult to interpret, and there are several competing techniques one can use to explain a search engine's behavior. We show that the theory of fair credit assignment provides a *unique* axiomatic solution that generalizes several existing recommendation- and metric-explainability techniques in the literature. Using this formalism, we show when existing approaches violate "fairness" and derive methods that sidestep these shortcomings and naturally handle counterfactual information. More specifically, we show existing approaches implicitly approximate second-order Shapley-Taylor indices and extend CAM, GradCAM, LIME, SHAP, SBSM, and other methods to search engines. These extensions can extract pairwise correspondences between images from trained *opaque-box* models. We also introduce a fast kernel-based method for estimating Shapley-Taylor indices that require orders of magnitude fewer function evaluations to converge. Finally, we show that these game-theoretic measures yield more consistent explanations for image similarity architectures.

## 1 Introduction

Search, recommendation, retrieval, and contrastive similarity learning powers many of today's machine learning systems. These systems help us organize information at scales that no human could match. The recent surge in million and billion parameter contrastive learning architectures for vision and language underscore the growing need to understand these classes of systems (Nayak, 2019; Chen et al., 2020b;a; Radford et al., 2021; Caron et al., 2020). Like classifiers and regressors, contrastive systems face a key challenge: richer models can improve performance but hinder interpretability. In high-risk domains like medicine, incorrect search results can have serious consequences. In other domains, search engine bias can disproportionately ans systematically hide certain voices (Mowshowitz & Kawaguchi, 2002; Diaz, 2008; Goldman, 2005).

Currently, there are several competing techniques to understand a similarity model's predictions (Zhu et al., 2019; Zheng et al., 2020; Dong et al.; Selvaraju et al., 2017; Vaswani et al., 2017). However, there is no agreed "best" method and no a formal theory describing an "optimal" search explanation method. We show that the theory of fair credit assignment provides a uniquely determined and axiomatically grounded approach for "explaining" a trained model's similarity judgements. In many cases, existing approaches are special cases of this formalism. This observation allows us to design variants of these methods that better satisfy the axioms of fair credit assignment and can handle counterfactual or relative explanations. Though we explore this topic through the lens of visual search, we note that these techniques could also apply to text, tabular, or audio search systems.

This work identifies two distinct classes of search engine explainability methods. "First order" approaches highlight the most important pixels that contribute to the similarity of objects and "Second order" explanations provide a full correspondence between the parts of query and retrieved image. We relate first order interpretations to existing theory on classifier explainability through a generic function transformation, as shown in the third column of Figure 1. We find that second order explanations correspond to a uniquely specified generalization of the Shapley values (Sundararajan et al., 2020) and is equivalent to projecting Harsanyi Dividends onto low-order subsets (Harsanyi, 1963). We use this formalism to create new second-order generalizations of Class Activation Maps (Zhou et al., 2016), GradCAM (Selvaraju et al., 2017), LIME (Ribeiro et al., 2016), and SHAP (Lundberg & Lee, 2017). Our contributions generalize several existing methods, illustrate a rich mathematical

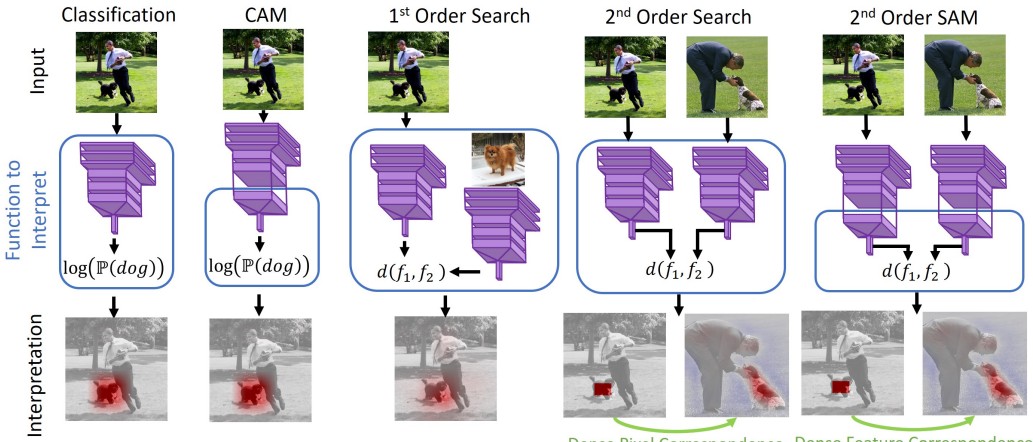

Figure 1: Architectures for search engine interpretability. Like classifier explanations, First-order search explanations yield heatmaps of important pixels for similarity (bottom row third column). Second order search interpretation methods yield a dense correspondence between image locations (last two columns). CAM (second column) is a particular case of Shapley value approximation, and we generalize it to yield dense correspondences (last column).

structure connecting model explainability and cooperative game theory, and allow practitioners to understand search engines with greater nuance and detail. We include a short video detailing the work at https://aka.ms/axiomatic-video. In summary we:

- Present the first uniquely specified axiomatic framework for *model-agnostic* search, retrieval, and metric learning interpretability using the theory of Harsanyi dividends.
- Show that our framework generalizes several existing model explanation methods (Zhou et al., 2016; Selvaraju et al., 2017; Zhu et al., 2019; Ribeiro et al., 2016) to yield dense pairwise correspondences between images and handle counterfactual information.
- Introduce a new kernel-based approximator for Shapley-Taylor indices that requires about $10\times$ fewer function evaluations.
- Show that our axiomatic approaches provide more faithful explanations of image similarity on the PascalVOC and MSCoCo datasets.

## 2 BACKGROUND

This work focuses on search, retrieval, metric learning, and recommendation architectures. Often, these systems use similarity between objects or learned features (Bengio et al., 2013) to rank, retrieve, or suggest content (Bing, 2017; Su & Khoshgoftaar, 2009; Chopra et al., 2005; Radford et al., 2021). More formally, we refer to systems that use a distance, relevance, or similarity function of the form: $d : \mathcal{X} \times \mathcal{Y} \to \mathbb{R}$ to quantify the relationship between items from sets $\mathcal{X}$ and $\mathcal{Y}$. In search and retrieval, $\mathcal{X}$ represents the space of search queries and $\mathcal{Y}$ represents the space of results, the function $d$ assigns a relevance to each query result pair. Without loss of generality, we consider $d$ as a "distance-like" function where smaller values indicate more relevance. The expression $\arg\min_{y \in \mathcal{Y}} d(x, y)$ yields the most relevant result for a query $x \in \mathcal{X}$.

Specializing this notion yields a variety of different kinds of ML systems. If $\mathcal{X} = \mathcal{Y} = \mathrm{Range}(\mathcal{N}(\cdot))$ where $\mathcal{N}$ is an image featurization network such as ResNet50 (He et al., 2016), the formalism yields a visual search engine or "reverse image search". Though this work focuses on visual search, we note that if $\mathcal{X}$ is the space of character sequences and $\mathcal{Y}$ is the space of webpages, this represents web search. In recommendation problems, $\mathcal{X}$ are users and $\mathcal{Y}$ are items, such as songs or news articles. In this work we aim to extract meaningful "interpretations" or "explanations" of the function $d$.

### 2.1 MODEL INTERPRETABILITY

The Bias-Variance trade-off (Kohavi et al., 1996) affects all machine learning systems and governs the relationship between a model's expressiveness and generalization ability. In data-rich scenarios, a model's bias dominates generalization error and increasing the size of the model class can improve performance. However, increasing model complexity can degrade model interpretability because

Figure 2: Comparison of first-order search interpretation methods which highlight pixels that contribute to similarity in red. Integrated Gradients (on pixels) struggles because well trained classifiers are invariant to minor pixel changes and have uninformative gradients.

added parameters can lose their connection to physically meaningful quantities. This affects not only classification and regression systems, but search and recommendation architectures as well. For example, the Netflix-prize-winning "BellKor" algorithm (Koren, 2009), boosts and ensembles several different methods making it difficult to interpret through model parameter inspection alone.

To tackle these challenges, some works introduce model classes that are naturally interpretable (Nori et al., 2019; Hastie & Tibshirani, 1990). Alternatively, other works propose *model-agnostic* methods to explain the predictions of classifiers and regressors. Many of these approaches explain the local structure around a specific prediction. Lundberg & Lee (2017) show that the Shapley value (Shapley, 1951), a measure of fair credit assignment, provides a unique and axiomatically characterized solution to classifier interpretability (SHAP). Furthermore, they show that Shapley values generalize LIME, DeepLIFT (Shrikumar et al., 2017), Layer-Wise Relevance Propagation (Bach et al., 2015), and several other methods (Štrumbelj & Kononenko, 2014; Datta et al., 2016; Lipovetsky & Conklin, 2001; Saabas, 2014). Many works in computer vision use an alternative approach called Class Activation Maps (CAMs). CAM projects the predicted class of a deep global average pooled (GAP) convolutional network onto the feature space to create a low resolution heatmap of class-specific network attention. GradCAM (Selvaraju et al., 2017) generalizes CAM to architectures other than GAP and can explain a prediction using only a single network evaluation. In Section 4 we show that CAM, GradCAM, and their analogue for search engine interpretability, Zhu et al. (2019), are also unified by the Shapley value and its second order generalization, the Shapley-Taylor index.

## 2.2 Fair Credit Assignment and the Shapley Value

Shapley values provide a principled and axiomatic framework for classifier interpretation. We briefly overview Shapley values and point readers to Molnar (2020) for more detail. Shapley values originated in cooperative game theory as the **only** fair way to allocate the profit of a company to its employees based on their contributions. To formalize this notion we define a "coalition game" as a set $N$ of $|N|$ players and a "value" function $v : 2^N \to \mathbb{R}$. In cooperative game theory, this function $v$ represents the expected payout earned by cooperating coalition of players. Shapley (1951) show that the unique, fair credit assignment to each player, $\phi_v(i \in N)$, can be calculated as:

$$\phi_v(i) := \sum_{S \subseteq N \setminus \{i\}} \frac{|S|!(|N| - |S| - 1)!}{|N|!} (v(S \cup \{i\}) - v(S)) \tag{1}$$

Informally, this equation measures the average increase in value that a player $i$ brings to a coalition $S$ by weighting each increase, $v(S \cup \{i\}) - v(S)$, by the number of ways this event could have happened during the formation of the "grand coalition" $N$. We note that this assignment, $\phi_v$, is the *unique* assignment that satisfies four reasonable properties: **symmetry** under player re-labeling, no credit assignment to **dummy** players, **linearity** (or it's alternative **monotonicity**), and **efficiency** which states that Shapley values should sum to $v(N) - v(\emptyset)$ (Young, 1985). Intuitively, these axioms require that a fair explanation should treat every feature equally (Symmetry), should not assign importance to features that are not used (Dummy), should behave linearly when the value function is transformed (Linear), and should sum to the function's value (Efficiency).

Shapley Values provide a principled way to explain the predictions of a machine learning model. To connect this work to model interpretability, we can identify the "features" used in a model as the "players" and interpret the value function, $v(S)$, as the expected prediction of the model when features $N \setminus S$ are replaced by values from a "background" distribution. This background distribution allows for "counterfactual" or relative explanations (Goyal et al., 2019).

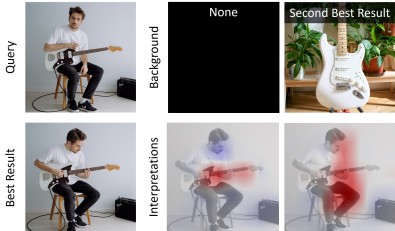 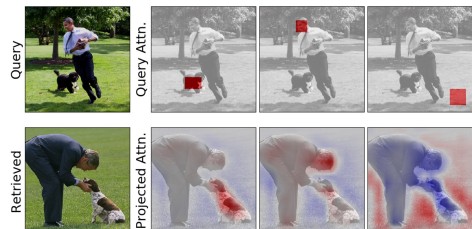

Figure 3: Explanations relative to a background distribution show why a result is better than an alternative. When asked why the best result (lower left) was better than the second best result (top right) our method correctly selects the player.

Figure 4: Visualization of how regions of two similar images "correspond" according to the second-order search interpretability method SAM. We can use this correspondence to transfer labels or attention between similar images.

## 3 RELATED WORK

There is a considerable body of literature on model interpretability and we mention just a handful of the works that are particularly related. One of our baseline methods, Dong et al., was one of the first to present a generic visual search engine explanation reminiscent of a Parzen-Window based estimator. Fong & Vedaldi (2017) introduce a method for explaining classifiers based on meaningful perturbation and Chefer et al. (2021) introduce a method for improving interpretation for transformer-based classifiers. Zhu et al. (2019) lifted CAM to search engines and we find that our Shapley-Taylor based method aligns with their approach for GAP architectures. Singh & Anand (2019) and Fernando et al. (2019) use LIME and DeepSHAP to provide first-order interpretations of text but do not apply their methods to images. Ancona et al. (2019) introduce a distribution propagation approach for improving the estimation of Shapley Values for deep models and can be combined with our approach. Many works implicitly use components that align with Shapley-Taylor indices for particular functions. Works such as Fischer et al. (2015a); Sun et al. (2020); Wang et al. (2020a); Chen et al. (2020c); Hou et al. (2019) use feature correlation layers to estimate and utilize correspondences between images. We show these layers are equivalent to Shapley-Taylor indices on the GAP architecture, and this allows create a correlation layer that handles counterfactual backgrounds. Other recent works have used learned co-attention within transformer architectures to help pool and share information across multiple domain types (Wei et al., 2020). Fu et al. (2020) attempt to learn a variant of GradCAM that better aligns with axioms similar to Shapley Values by adding efficiency regularizers. The method is not guaranteed to satisfy the axioms but is more "efficient".

We rely on several works to extend Shapley values to more complex interactions. Harsanyi (1963) generalized the Shapely value by introducing a "dividend" that, when split and distributed among players, yields the Shapley values. Owen (1972) introduces an equivalent way to extend Shapley values using a multi-linear extension of the game's characteristic function. Sundararajan et al. (2020) introduce the Shapley-Taylor index and show is equivalent to the Lagrangian remainder of Owen's multi-linear extension. Integrated Hessians (Janizek et al., 2020) enable estimation of a second-order variant of the Aumann-Shapley values and we use this approach to create a more principled second-order interpretation method for differentiable search engines.

## 4 UNIFYING FIRST-ORDER SEARCH INTERPRETATION TECHNIQUES

Though there is a considerable body of work on opaque-box classifier interpretability, opaque-box search engine interpretability has only recently been investigated (Singh & Anand, 2019; Zhu et al., 2019; Zheng et al., 2020). We introduce an approach to transform opaque and grey-box classification explainers into search engine explainers, allowing us to build on the rich body of existing work for classifiers. More formally, given a similarity function $d : \mathcal{X} \times \mathcal{Y} \to \mathbb{R}$ and elements $x \in \mathcal{X}$ and $y \in \mathcal{Y}$ we can find the "parts" of $y$ that most contribute to the similarity by computing the Shapley values for the following value function:

$$v_1(S) : 2^N \to \mathbb{R} := d(x, mask(y, S)) \tag{2}$$

Where the function $mask(\cdot, \cdot) : \mathcal{Y} \times 2^N \to \mathcal{Y}$, replaces "parts" of $y$ indexed by $S$ with components from a background distribution. Depending on the method, "parts" could refer to image superpixels, small crops, or locations in a deep feature map. This formula allows us to lift many existing approaches to search engine interpretability. For example, let $\mathcal{X}$, and $\mathcal{Y}$ represent the space of pixel

representations of images. Let the grand coalition, $N$, index a collection of superpixels from the retrieved image $y$. Let $mask(y, S)$ act on an image $y$ by replacing the $S$ superpixels with background signal. With these choices, the formalism provides a search-engine specific version of ImageLIME and KernelSHAP. Here, Shapley values for each $i \in S$ measure the impact of the corresponding superpixel on the similarity function. If we replace superpixels with hierarchical squares of pixels we arrive at Partition SHAP (Lundberg). We can also switch the order of the arguments to get an approach for explaining the query image's impact on the similarity. In Figure 2 we qualitatively compare how methods derived from our approach compare to two existing approaches: SBSM (Dong et al.) and VESM (Zheng et al., 2020), on a pair of images and a MocoV2 based image similarity model. In addition to generalizing LIME and SHAP we note that this approach generalizes VEDML (Zhu et al., 2019), a metric-learning adaptation of CAM:

**Proposition 4.1** *Let $\mathcal{X} = \mathcal{Y} = \mathbb{R}^{CHW}$ and represent the space of deep network features where $C, H, W$ represent a channel, height, and width of the feature maps respectively. Let the function $d := \sum_c GAP(x)_c GAP(y)_c$. Let the grand coalition, $N = [0, H] \times [0, W]$, index the spatial coordinates of the image feature map $y$. Let the function $mask(y, S)$ act on a feature map $y$ by replacing the features at locations $S$ with a background signal $b$. Then:*

$$\phi_{v_1}((h, w) \in N) = \frac{1}{HW} \sum_c GAP(x)_c (y_{chw} - b_{chw}) \tag{3}$$

Where GAP refers to global average pooling. We defer proof of this and other propositions to the Supplement. The results of this proposition mirrors the form of VEDML but with an added term to handle background distributions. These extra terms broaden the applicability of VEDML and we demonstrate their effect on explanations in Figure 3. In particular, we explain why two guitar players are similar in general (no background distribution), and relative to the second-best result of a guitar. Without a background, the explanation focuses on the guitar. However, when the explanation is relative to an image of a guitar the explanation focuses instead on the "tie-breaking" similarities, like the matching player. With counterfactual queries one can better understand a model's rationale behind *relative* similarity judgements and this can help in domains such as search engine optimization and automated medical diagnosis. We refer to Equation 3 as the Search Activation Map (SAM) in analogy with the Class Activation Map. We note that in non-GAP architectures, VEDML requires Taylor approximating nonlinear components. This heuristic corresponds estimating the Shapley values for a linear approximation of the true value function. For nonlinear architectures such as those that use cosine similarity, SAM diverges from Shapley value theory and hence violates its axioms. We can remedy this by using a Kernel-based Shapley value approximator (Lundberg & Lee, 2017) and refer to this approach as Kernel SAM.

Though the Shapley value framework unifies several methods for search engine interpretability, we note that the popular technique GradCAM does not align with Shapley value theory when applied to our feature-based value function (though it does align with Shapley values for GAP *classifiers*). To connect this approach to the theory of fair credit assignment, we show that GradCAM closely resembles Integrated Gradients (IG) (Sundararajan et al., 2017b), an approximator to the Aumann-Shapley values (Aumann & Shapley, 2015):

**Proposition 4.2** *Let $v(S) : [0, 1]^N \to \mathbb{R} := f(mask(x, S))$ represent soft masking of the spatial locations of a deep feature map $x$ with the vector of zeros and applying a differentiable function $f$. GradCAM is equivalent to Integrated Gradients approximated with a single sample at $\alpha = 1$ only if the function $f$ has spatially invariant derivatives:*

$$\forall (h, w), (i, j) \in N : \frac{\partial f(x)}{\partial x_{chw}} = \frac{\partial f(x)}{\partial x_{cij}}$$

*In typical case where $f$ does not have spatially invariant derivatives GradCAM violates the dummy axiom (see Section 2.2) and does not represent an approximation of Integrated Gradients.*

Where $\alpha$ refers to the parameter of IG that blends background and foreground samples. We note that the Aumann-Shapley values generalize the Shapley value to games where infinite numbers of players can join finitely many "coalitions". These values align with Shapley values for linear functions but diverge in the nonlinear case. Proposition 4.2 also shows that in general GradCAM is sub-optimal and can be improved by considering Integrated Gradients on the feature space. We refer to this modification to GradCAM as Integrated Gradient Search Activation Maps or "IG SAM". We also note that this modification can be applied to classifier-based GradCAM to yield a more principled

classifier interpretation approach. We explore this and show an example of GradCAM violating the dummy axiom in the Supplement.

## 5 SECOND-ORDER SEARCH INTERPRETATIONS

Visualizing the pixels that explain a similarity judgement provides a simple way to inspect where a retrieval system is attending to. However, this visualization is only part of the story. Images can be similar for many different reasons, and a good explanation should clearly delineate these independent reasons. For example, consider the pair of images in the left column of Figure 6. These images show two similar scenes of people playing with dogs, but in different arrangements. We seek not just a heatmap highlighting similar aspects, but a data-structure capturing how parts of the query image *correspond* to parts of a retrieved image. To this end we seek to measure the interaction strength between areas of query and retrieved images as opposed to the effect of single features. We refer to this class of search and retrieval explanation methods as "second-order" methods due to their relation with second-order terms in the Shapley-Taylor expansion in Section 5.1.

### 5.1 HARSANYI DIVIDENDS

To capture the notion of interactions between query and retrieved images, we must consider credit assignments to *coalitions* of features. (Harsanyi, 1963) formalize this notion with a unique and axiomatically specified way to assign credit or "Harsanyi Dividends" to every possible coalition, $S$, of $N$ players in a cooperative game using the formula:

$$d_v(S) := \begin{cases} v(S) & \text{if } |S| = 1 \\ v(S) - \sum_{T \subsetneq S} d_v(T) & \text{if } |S| > 1 \end{cases} \tag{4}$$

These dividends provide a detailed view of the function's behavior at every coalition. In particular, Harsanyi (1963) show that Shapley values arise from distributing these dividends evenly across members of the coalitions, a process we refer to a "projecting" the dividends down. In this work we seek a second-order analog of the Shapley values, so we generalize the notion of sharing these dividends between individuals to sharing these dividends between sub-coalitions. This computation re-derives the recently proposed Shapley-Taylor Indices (Sundararajan et al., 2020), which generalize the Shapley values to coalitions of a size $k$ using the discrete derivative operator. More specifically, by sharing dividends, we can alternatively express Shapley-Taylor values for coalitions $|S| = k$ as:

$$\phi_v^k(S) = \sum_{T : S \subset T} \frac{d_v(T)}{\binom{|T|}{|S|}} \tag{5}$$

Which states that the Shapley-Taylor indices arise from projecting Harsanyi dividends onto the $k^{th}$ order terms. We note that this interpretation of the Shapley-Taylor indices is slightly more flexible than that of Sundararajan et al. (2020) as it allows one to define "jagged" fair credit assignments over just the coalitions of interest. Equipped with the Shapley-Taylor indices, $\phi_v^k$, we can now formulate a value function for "second-order" search interpretations. As in the first order case, consider two spaces $\mathcal{X}, \mathcal{Y}$ equipped with a similarity function $d$. We introduce the second-order value function:

$$v_2(S) : 2^N \to \mathbb{R} := d(mask(x, S), mask(y, S)) \tag{6}$$

Where the grand coalition, $N = L_q \cup L_r$, are "locations" in both the query and retrieved images. These "locations" can represent either superpixels or coordinates in a deep feature map. Our challenge now reduces to computing Shapley-Taylor indices for this function.

### 5.2 A FAST SHAPLEY-TAYLOR APPROXIMATION KERNEL

Though the Harsanyi Dividends and Shapley-Taylor indices provide a robust way to allocate credit, they are difficult to compute. The authors of the Shapley-Taylor indices provide a sampling-based approximation, but this requires estimating each interaction term separately and scales poorly as dimensionality increases. To make this approach more tractable for high dimensional functions we draw a parallel to the unification of LIME with Shapley values through a linear regression weighting kernel. In particular, one can efficiently approximate Shapley values by randomly sampling coalitions, evaluating the value function, and fitting a weighted linear map from coalition vectors to function values. We find that this connection between Shapley values and weighted linear models naturally lifts to a weighted quadratic estimation problem in the "second-order" case. In particular, we introduce a weighting kernel for second order Shapley-Taylor indices:

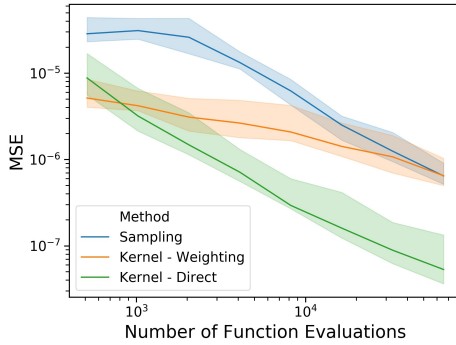

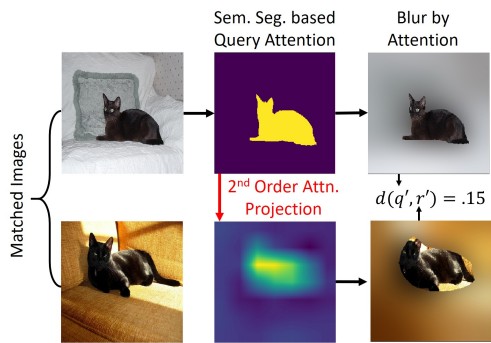

Figure 5: Convergence of Shapley-Taylor estimation schemes with respect to the Mean Squared Error (MSE) on randomly initialized deep networks with 15 dimensional input. Our strategies (Kernel) converge with significantly fewer function evaluations.

Figure 6: Our Second-order explanation evaluation strategy. A good method should project query objects (top left and middle) to corresponding objects in the retrieved image (bottom left and middle). When censoring all but these shared objects (right column) the search engine should view these images as similar.

$$\Lambda(S) = \frac{|N| - 1}{\binom{|N|}{|S|}\binom{|S|}{2}(|N| - |S|)} \tag{7}$$

Using this kernel, one can instead sample random coalitions, evaluate $v$, and aggregate the information into weighted quadratic model with a term for each distinct coalition $|S| \leq 2$. This allows one to approximate *all* Shapley-Taylor indices of $k = 2$ with a single sampling procedure, and often requires $10\times$ fewer function evaluations to achieve the same estimation accuracy. We show this speedup in Figure 5 on randomly initialized 15-dimensional deep networks. A detailed description of this and other experiments in this work are in the supplement. We find that one can further speed up the method by directly sampling from the induced distribution (Kernel-Direct) as opposed to randomly sampling coalitions and calculating weights (Kernel-Weighting). This direct sampling can be achieved by first sampling the size of the coalition from $p(s) \propto (|N| - 1)/(\binom{s}{2}(|N| - s))$ and then randomly sampling a coalition of that size. When our masking function operates on super-pixels, we refer to this as the second-order generalization of Kernel SHAP. This also gives insight into the proper form for a second-order generalization of LIME. In particular we add L1 regularization (Tibshirani, 1996) and replace our kernel with a local similarity, $\Lambda(S) = \exp(-\lambda|mask(x, S); mask(y, S) - x; y|_2^2)$ where ";" represents concatenation, to create a higher-order analogue of LIME. Finally we note that certain terms of the kernel are undefined due to the presence of $\binom{s}{2}$ and $|N| - |S|$ in the denominator. These "infinite" weight terms encode hard constraints in the linear system and correspond to the efficiency axiom. In practice we enumerate these terms and give them a very large weight ($10^8$) in our regression. We reiterate that our kernel approximator converges to the same, uniquely-defined, values as prior sampling approaches but requires significantly fewer function evaluations.

## 5.3 SECOND-ORDER SEARCH ACTIVATION MAPS

In the first-order case, CAM and its search engine generalization, Search Activation Maps, arise naturally from the Shapley values of our first-order value function, Equation 2. To derive a second order generalization of SAM we now look to the Shapley-Taylor indices of our second order value function, Equation 6, applied to the same GAP architecture described in Proposition 4.1.

**Proposition 5.1** *Let the spaces $\mathcal{X}$, $\mathcal{Y}$ and function $d$ be as in Proposition 4.1. Let the grand coalition, $N$, index into the spatial coordinates of both the query image features $x$ and retrieved image features $y$. Let the function $mask(y, S)$ act on a feature map $y$ by replacing the corresponding features with a background feature map $a$ for query features and $b$ for retrieved features. Then:*

$$\phi_{v_2}(\{(h, w) \in \mathcal{L}_q, (i, j) \in \mathcal{L}_r\}) = \frac{1}{H^2 W^2} \sum_c x_{chw} y_{cij} - a_{chw} y_{cij} - x_{chw} b_{cij} + a_{chw} b_{cij} \tag{8}$$

We note that the first term of the summation corresponds to the frequently used correlation layer (Fischer et al., 2015b; Sun et al., 2020; Wang et al., 2020a; Chen et al., 2020c) and generalizes

Table 1: Comparison of performance of first- and second-order search explanation methods. Methods introduced in this work are highlighted in pink. *Though SAM generalizes (Zhu et al., 2019) we refer to it as a baseline. For additional details see Section 6

| Metric | Order | Model | SBSM | PSHAP | LIME | KSHAP | VESM | GCAM | SAM* | IG SAM | KSAM |
|---|---|---|---|---|---|---|---|---|---|---|---|
| | | | Model Agnostic | | | | Architecture Dependent | | | | |
| Faithfulness | First | DN121 | 0.18 | **0.26** | 0.23 | 0.24 | 0.08 | 0.12 | 0.12 | **0.20** | **0.20** |
| | | MoCoV2 | 0.22 | **0.30** | 0.28 | **0.30** | 0.13 | 0.19 | 0.21 | 0.25 | **0.25** |
| | | RN50 | 0.11 | **0.16** | 0.14 | 0.14 | 0.04 | 0.08 | 0.07 | **0.11** | **0.11** |
| | | VGG11 | 0.14 | **0.16** | 0.15 | 0.15 | 0.05 | 0.09 | 0.11 | **0.14** | **0.14** |
| | Second | DN121 | 0.48 | - | **0.54** | **0.54** | - | - | 0.48 | 0.48 | **0.49** |
| | | MoCoV2 | 0.69 | - | **0.74** | **0.74** | - | - | **0.72** | 0.70 | 0.71 |
| | | RN50 | 0.74 | - | **0.77** | **0.77** | - | - | **0.74** | 0.74 | 0.74 |
| | | VGG11 | 0.68 | - | **0.71** | **0.71** | - | - | 0.69 | 0.69 | **0.70** |
| Inefficiency | First | DN121 | - | **0.00** | 0.20 | **0.00** | - | 12.8 | 0.56 | 0.02 | **0.00** |
| | | MoCoV2 | - | **0.00** | 0.10 | **0.00** | - | 0.46 | 0.53 | 0.03 | **0.00** |
| | | RN50 | - | **0.00** | 0.22 | **0.00** | - | 14.9 | 0.47 | 0.03 | **0.00** |
| | | VGG11 | - | **0.00** | 0.27 | **0.00** | - | 4.20 | 0.54 | 0.05 | **0.00** |
| | Second | DN121 | - | - | 0.14 | **0.01** | - | - | 0.21 | 0.03 | **0.01** |
| | | MoCoV2 | - | - | 0.13 | **0.01** | - | - | 0.20 | 0.02 | **0.01** |
| | | RN50 | - | - | 0.06 | **0.01** | - | - | 0.06 | **0.01** | **0.01** |
| | | VGG11 | - | - | 0.11 | **0.01** | - | - | 0.22 | 0.03 | **0.01** |
| mIoU | Second | DN121 | 0.55 | - | **0.68** | 0.67 | - | - | **0.68** | **0.68** | 0.67 |
| | | MoCoV2 | 0.57 | - | **0.70** | 0.69 | - | - | **0.70** | **0.70** | 0.69 |
| | | RN50 | 0.55 | - | **0.67** | 0.66 | - | - | **0.69** | 0.66 | 0.65 |
| | | VGG11 | 0.54 | - | **0.68** | 0.67 | - | - | 0.72 | **0.73** | 0.70 |

the "point-to-point" signal in Zhu et al. (2019). In particular, our axiomatically derived version has the extra terms allow counterfactual explanations against different background signals. Like in the first-order case, this closed form only holds in the GAP architecture. To extend the method in a principled way we use our second-order kernel approximator and refer to this as second-order KSAM. We also introduce a generalization using a higher order analogue of Integrated Gradients, Integrated Hessians (Janizek et al., 2020), applied to our feature maps. We refer to this as second-order IGSAM. In Section A.3 of the Supplement we prove that this approach is proportional to the Shapley-Taylor indices for the GAP architecture. We can visualize these second-order explanations by aggregating these Shapley-Taylor indices into a matrix with query image locations as rows and retrieved locations as columns. Using this matrix, we can "project" signals from a query to retrieved image. We show a few examples of attention projection using our second-order SAM in Figure 4.

## 6 EXPERIMENTAL EVALUATION

**First Order Evaluation** Evaluating the quality of an interpretability method requires careful experimental design and is independent from what "looks good" to our human eye. If a model explanation method produces "semantic" connections between images it should be because to the underlying model is sensitive to these semantics. As a result, we adopt the evaluation strategy of Lundberg & Lee (2017), which measures how well the model explanation approximates the expected influence of individual features. In particular, these works calculate each feature's importance, replace the top $n\%$ of features with background signal, and measure the effect on the function. A good model interpretability method should cause the replacement of the most important features, and hence cause the largest expected change in the function. We refer to this metric as the "Faithfulness" of an interpretation measure as it directly measures how well an interpretation method captures the behavior of an underlying model. Figure 7 in the Supplement diagrams this process for clarity. In our experiments we blur the top $30\%$ of image pixels to compute faithfulness. For those methods that permit it, we also measure how much the explanation violates the efficiency axiom. In particular we compare the sum of explanation coefficients with the value of $v(N) - v(\emptyset)$ and refer to this as the "Inefficiency" of the method. For additional details and evaluation code please see Section A.2 in the Supplement.

**Second Order Evaluation**   In the second-order case we adopt the evaluation strategy of Janizek et al. (2020) which introduce a analogous second-order faithfulness measure. In particular, we measure how well model explanations approximate the expected *interaction* between two features. To achieve this, we select an object from the query image, use the second order explanation to find the corresponding object in the retrieved image, censor all but these two objects. We measure the new similarity as a measure of Faithfulness and illustrate this process in In Figure 6. We additionally quantify the inefficiency of several second-order methods as well as their effectiveness for semantic segmentation label propagation. In particular, we measure how well the explanation method can project a source object onto a target object. We treat this as a binary segmentation problem and measure the mean intersection over union (mIoU) of the projected object with respect to the true object mask. We note that mIoU is not a direct measurement of interpretation quality, but it can be useful for those intending to use model-interpretation methods for label propagation (Ahn et al., 2019; Wang et al., 2020b). These results demonstrate that axiomatically grounded model explanation methods such as IG SAM could offer improvement on downstream tasks. Because human evaluations introduce biases such as preference for compact or smoothness explanations, we consider Mechanical Turk (Paolacci et al., 2010) studies outside the scope of this work.

**Datasets**   We evaluate our methods on the Pascal VOC (Everingham et al., 2010) and MSCoCo Caesar et al. (2018) semantic segmentation datasets. To compute first and second order faithfulness we mine pairs of related images with shared object classes. We use the MoCo V2 (Chen et al., 2020b) unsupervised image representation method to featurize the training and validation sets. For each image in the validation set we choose a random object from the image and find the training image that contains an object of the same class (Hamilton et al., 2020).

**Results**   In Table 1 and Table 4 of the Supplement we report experimental results for PascalVOC and MSCoCo respectively. We evaluate across visual search engines created from four different backbone networks: DenseNet121 (Huang et al., 2017), MoCo v2 (Chen et al., 2020b), ResNet50 (He et al., 2016), and VGG11 (Simonyan & Zisserman, 2014) using cosine similarity on GAP features. As baselines we include VESM, SBSM, and SAM which generalizes (Zhu et al., 2019). We note that SBSM was not originally presented as a second-order method, and we describe how it can be lifted to this higher order setting in Section A.11 of the Supplement. We also evaluate several existing classifier explanation approaches applied to our search explanation value functions such as Integrated Gradients (Sundararajan et al., 2017a) on image pixels, Partition SHAP (Lundberg), LIME, Kernel SHAP (KSHAP), and GradCAM (GCAM) on deep feature maps (Selvaraju et al., 2017). For second-order variants of LIME and SHAP we used the local weighting kernel and our Shapley-Taylor approximation kernel from Section 5.2. Overall, several key trends appear. First, Shapley and Aumann-Shapley based approaches tend to be the most faithful and efficient methods, but at the price of longer computation time. One method that strikes a balance between speed and quality is our Integrated Gradient generalization of CAM which has both high faithfulness, low inefficiency, and only requires a handful of network evaluations ($\sim 10^2$). Furthermore, grey-box feature interpretation methods like SAM and IG SAM tend to perform better for label propagation. Finally, our methods beat existing baselines in several different categories and help to complete the space of higher order interpretation approaches. We point readers to the Section A.2 for additional details, compute information, and code.

## 7   CONCLUSION

In this work we have presented a uniquely specified and axiomatic framework for *model-agnostic* search, retrieval, and metric learning interpretability using the theory of Harsanyi dividends. We characterize search engine interpretability methods as either "first" or "second" order methods depending on whether they extract the most important areas or pairwise correspondences, respectively. We show that Shapley values of a particular class of value functions generalize many first-order methods, and this allows us to fix issues present in existing approaches and extend these approaches to counterfactual explanations. For second order methods we show that Shapley-Taylor indices generalize the work of Zhu et al. (2019) and use our framework to introduce generalizations of LIME, SHAP, and GradCAM. We apply these methods to extract image correspondences from opaque-box similarity models, a feat not yet presented in the literature. To accelerate estimation higher order Shapley-Taylor indices, we contribute a new weighting kernel that requires $10\times$ fewer function evaluations. Finally, we show this game-theoretic formalism yields methods that are more "faithful" to the underlying model and better satisfy efficiency axioms across several visual similarity methods.

ACKNOWLEDGMENTS

We would like to thank Siddhartha Sen for sponsoring access to the Microsoft Research compute infrastructure. We would also like to thank Zhoutong Zhang, Jason Wang, and Markus Weimer for their helpful commentary on the work. We thank the Systems that Learn program at MIT CSAIL for their funding and support.

This material is based upon work supported by the National Science Foundation Graduate Research Fellowship under Grant No. 2021323067. Any opinion, findings, and conclusions or recommendations expressed in this material are those of the authors(s) and do not necessarily reflect the views of the National Science Foundation.This work is supported by the National Science Foundation under Cooperative Agreement PHY-2019786 (The NSF AI Institute for Artificial Intelligence and Fundamental Interactions, http://iaifi.org/)

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

# A    APPENDIX

## A.1    VIDEO AND CODE

We include a short video description of our work at https://aka.ms/axiomatic-vdieo.

We also provide training and evaluation code at https://aka.ms/axiomatic-code

## A.2    EVALUATION AND IMPLEMENTATION DETAILS

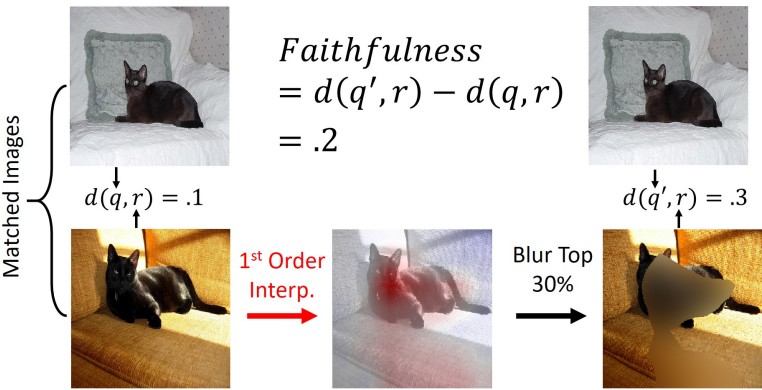

Figure 7: First-order interpretation evaluation strategy. A good method should highlight pixels in the query image (top left and middle) that, when censored (top right), have the largest possible impact on the cosine distance.

**Models:**    Our evaluation experiments use visual similarity systems built from "backbone" networks that featurize images and compare their similarity using cosine distance. We consider supervised backbones and contrastive unsupervised backbones. In particular, ResNet50 (He et al., 2016), VGG11 (Simonyan & Zisserman, 2014), and DenseNet121 (Huang et al., 2017) are trained with human classification annotations from the ImageNet dataset (Deng et al., 2009) and MoCo V2 is trained using unsupervised contrastive learning on ImageNet. We use torchvision (Marcel & Rodriguez, 2010) based model implementations and pre-trained weights except for MocoV2 which we download from He & Wu (2021) (800 epoch model). For kernel convergence experiments in Figure 5 we use randomly initialized three layer deep networks with Glorot (Glorot et al., 2011) initialization, rectified linear unit activations, and a 20 dimensional hidden layer. We note that the functional form is not of much importance for these experiments so long as the function is nonlinear and non-quadratic. We provide an additional example in Figure 10 on random 15 dimensional Boolean functions formed by enumerating and summing all possible variable products and weighting each by a uniform coefficient between 0 and 10.

**Data:**    For evaluations within Table 1 we use the Pascal VOC (Everingham et al., 2010) dataset. In particular we form a paired image dataset by using MoCo V2 to featurize the training and validation sets. All experiments use images that have been bi-linearly resized to $224 \times 224$ pixels. For each image in the PascalVOC validation set we choose a random segmentation class that contains over $5\%$ of image pixels. We then find each validation image's closest "Conditional Nearest Neighbor" (Hamilton et al., 2020) from the images of the training set of the chosen segmentation class. We use cosine similarity between MoCoV2 deep features to find nearest neighbors. With this dataset of pairs, we can then compute our first and second order evaluation metrics. We provide instructions for downloading the pairs of images in the attached code. We note that our approach for selecting pairs of images with matching segmentation labels allow for measuring Faithfulness and success in label propagation as measured by mIoU.

**Metrics:** Our attached code contains implementations all metrics for preciseness but we include descriptions of metrics here for clarity. To measure first order faithfulness, we take a given validation image and training image from our dataset of paired images and compute the first order heat-map over the validation image. We then blur the top $30\%$ of pixels by blurring the image with a $25 \times 25$ pixel blur kernel and replacing the top $30\%$ of original image pixels with those from the blurred image. The drop in cosine similarity between the unblurred images and the unblurred training and blurred validation image is the first order faithfulness. We illustrate our first-order evaluation strategy in Figure 7.

For our second-order evaluation, we use the ground truth semantic segmentation mask of the training image as a "query" attention signal. We then use the second-order interpretation methods to "project" this attention to the "retrieved" validation image. We censor all but the most-attended pixels in the retrieved image. The size of the remaining pixels matches the size of the validation image's selected semantic segmentation mask. In the second-order case we additionally measure the mean intersection over union (mIoU) of the resulting mask compared to the ground-truth retrieved image segmentation. A good approach should attend to jut the pixels of the segmentation class and thus yield a mIoU of 1 (maximum value) as a binary segmentation problem. We illustrate our second-order evaluation strategy in Figure 6.

Finally, for those methods that permit it, we measure how much they violate the efficiency axiom by summing the interpretation coefficients and comparing with $v(N) - v(\emptyset)$. In the first order setting $v(N)$ is the similarity between query and retrieved image, and $v(\emptyset)$ is the similarity between query and a blurred retrieved image (with 25 pixel blur). In the second order setting $v(\emptyset)$ represents the similarity when both images are blurred. For SAM-based methods we replace features with those from blurred images. To compute the sum of interpretation coefficients for kernel methods we sum over Shapley values in the first order case and over Shapley-Taylor indices of order $k \leq 2$ in the second-order case. For Partition SHAP Lundberg we sum coefficients over all pixels. For Integrated Hessian's we sum over all first and second order coefficients as described in Janizek et al. (2020).

In tables we report mean values of Inefficiency, and Faithfulness metrics and note that for these experiemtns the Standard Error of the Mean (SEM) is far below the three significant figure precision of the table.

**First Order Methods:** For first order explanations we use the official implementation of Image-LIME (Ribeiro, 2021) and use the SHAP package for Integrated Gradients, Partition SHAP, and Kernel SHAP (Lundberg). We re-implement SBSM and VESM in PyTorch from the instructions provided in their papers. For sampling procedures such as LIME, Kernel SHAP, and Partition SHAP we use 5000 function evaluations. For first and second-order super-pixel based methods (LIME, Kernel-SHAP) we use the SLIC superpixel method (Achanta et al., 2010) provided in the Scipy library (Virtanen et al., 2020) with 50 segments, $compactness = 10$, and $\sigma = 3$. For SBSM we use a window size of 20 pixels and a stride of 3 pixels. We batch function evaluations with minibatch size 64 for backbone networks and $64 \times 20$ for SAM based methods. For all background distributions we blur the images with a 25-pixel blur kernel with the exception of LIME and SBSM which use mean color backgrounds.

**Second Order Methods:** For second order methods we use the same background and superpixel algorithms, but implement all methods within PyTorch for uniform comparison. For SBSM, Kernel SHAP, and LIME we use 20000 samples and for KSAM and IGSAM we use 40000 samples. For IGSAM we use the expected Hessians method referenced in the supplement of Janizek et al. (2020). We use the PyTorch "lstsq"function for solving linear systems. For more details on our generalization of SBSM see Section A.11.

**Compute and Environment:** Experiments use PyTorch (Paszke et al., 2019) v1.7 pre-trained models, on an Ubuntu 16.04 Azure NV24 Virtual Machine with Python 3.6. For all methods that require many network evaluations we use PyTorch DataLoaders with 18 background processes to eliminate IO bottlenecks. We standardize experiments using Azure Machine Learning and run each experiment on a separate virtual machine to avoid slowdowns due to scarce CPU or GPU resources.

## A.3   PROOF OF PROPOSITION 4.2

Let $v(S) : [0,1]^N \to \mathbb{R} := f(mask(x,S))$ represent soft masking of the spatial locations of a deep feature map $x$ with the vector of zeros and applying a differentiable function $f$. We begin with the formulation of Integrated Gradients:

$$\text{IG}_{hw}(S) = (S_{hw} - S'_{hw}) \int_{\alpha=0}^1 \frac{\partial v(\alpha S + (1-\alpha)S')}{\partial T_{hw}}$$

In our case the foreground, $S := \mathbb{1}^{HW}$, is a mask of all 1s and the background, $S'$, is the zero mask of the same shape. We note that the $\frac{\partial}{\partial T_{hw}}$ refers to taking the partial of the full input $\alpha S$, not just the mask $S$. We include this to stress the subtle difference which can be missed in a quick reading of the equations of Sundararajan et al. (2017a). In this case our formula is simplified to:

$$\text{IG}_{hw}(S) = \int_{\alpha=0}^1 \frac{\partial v(\alpha S)}{\partial T_{hw}}$$

Approximating this integral with a single sample at $\alpha = 1$ yields:

$$\begin{aligned}
\text{IG}_{hw}(S) &\approx \frac{\partial v(S)}{\partial S_{hw}} \\
&= \frac{\partial f(mask(x,S))}{\partial T_{hw}} \\
&= \frac{\partial}{\partial T_{hw}} f(x \odot S) \\
&= \sum_c x_{chw} \frac{\partial f(x)}{\partial x_{chw}} \\
&= \sum_c x_{chw} GAP(\nabla_x f(x)) \qquad \text{(Spatially Invariant Derivatives)}
\end{aligned}$$

Which is precisely the formulation of GradCAM. This also makes it clear that the global average pooling of GradCAM causes the method to deviate from integrated gradients in the general case. To construct a function where GradCAM violates the dummy axiom we simply have to violate the spatial invariance of gradients. We provide a specific example of this violation in A.4.

## A.4   GRADCAM VIOLATES THE DUMMY AXIOM

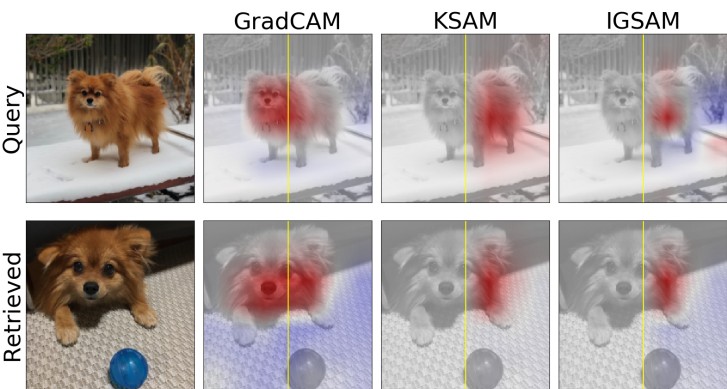

Figure 8: Interpretations of a function that purposely ignores the left half of the image. KSAM and IGSAM properly assign zero weight to these features. GradCAM does not and hence violates the dummy axiom of fair credit assignment.

It is straightforward to construct examples where GradCAM violates the dummy axiom. For example, consider the function:

$$d(x,y) = sim_{cosine}(GAP(x), GAP(y \odot M))$$

Where $sim_{cosine}$ represents cosine similarity, $\odot$ represents elementwise multiplcation, and $M \in [0,1]^{CHW}$ is a mask where $M_{chw} = 0$ if $w \leq \frac{W}{2}$ and $M_{chw} = 1$ otherwise. Intuitively, $M$ removes the influence of any feature on the left of the image making these features "dummy" features for the model. Because GradCAM spatially averages the gradients prior to taking the inner product with the feature map all features are treated equally regardless of how they are used. In this example, depicted in Figure 8, positive contributions from the right side of the image are extended to the left side of the image despite the fact that the mask, $M$ stops these features from impacting the prediction. Using a Shapley or Aumann-Shapley approximator on the feature space does not suffer from this effect as shown in the two right columns of Figure 8.

## A.5 INTEGRATED GRADIENT CAM

Sections A.4 and A.3 demonstrate that GradCAM can violate the dummy axiom when the function has spatially varying gradients which is a common occurrence especially if one is trying to interpret deeper layers of a network. We remedy this by instead considering Integrated Gradients on a function which masks the spatial locations of a deep feature map. More specifically our Integrated Gradient generalization of CAM takes the following form:

$$IGCAM(h, w) := \int_{\alpha=0}^{1} \frac{\partial f(b + \alpha M \odot (x - b))}{\partial T_{hw}} \tag{9}$$

Where $f$ is the classification "head", $x \in \mathbb{R}^{CHW}$ is a tensor of deep image features, $M := \mathbb{1}^{HW}$ is a mask of 1s over the spatial location of the features, $b \in \mathbb{R}^{CHW}$ is a background signal commonly taken to be zero in GradCAM. We note that the $\frac{\partial}{\partial T_{hw}}$ refers to taking the partial of the full input $b + \alpha M \odot (x - b)$, not just the mask. We include this to stress the subtle difference which can be missed in a quick reading of the equations of Sundararajan et al. (2017a). This variant of GradCAM does not violate the dummy axiom and satisfies the axioms of the Aumann-Shapley fair credit assignment.

## A.6 ADDITIONAL SIMILARITY VISUALIZATIONS

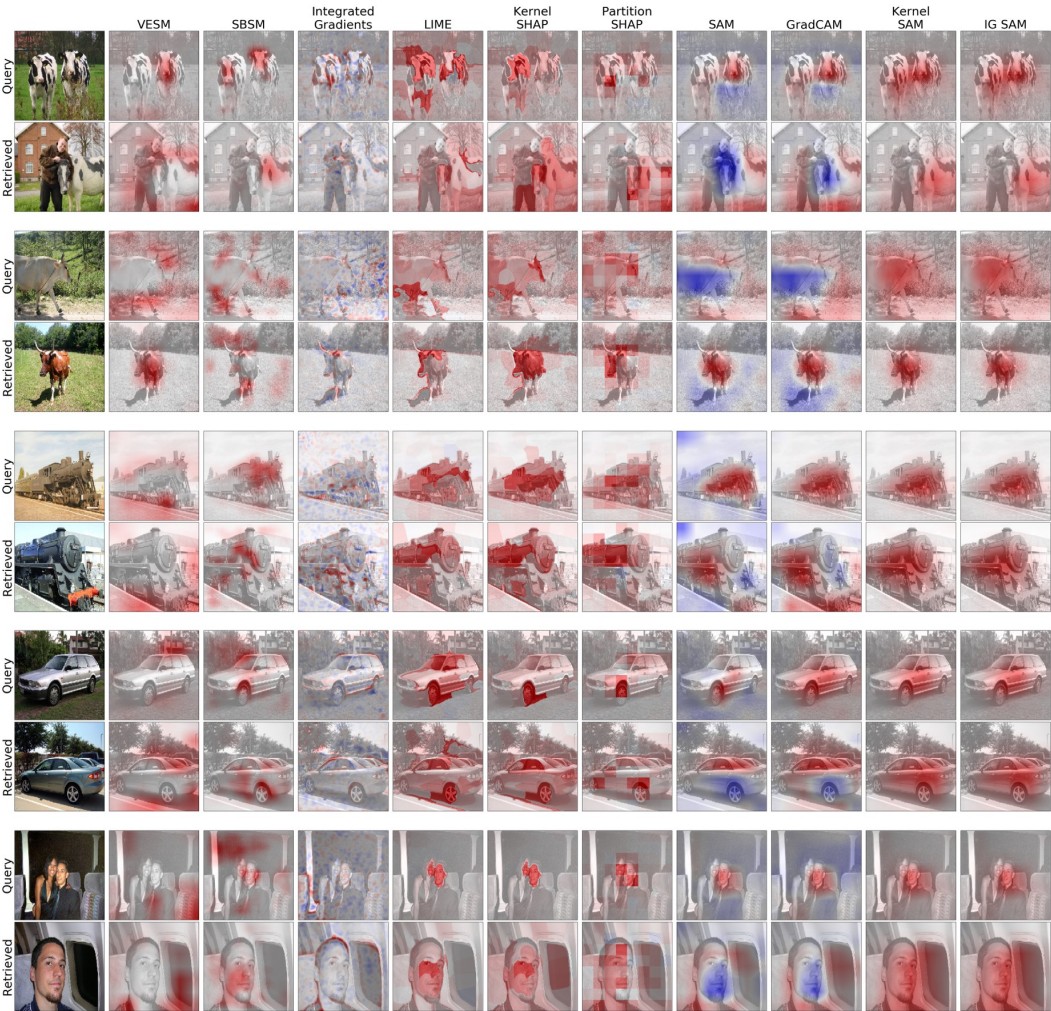

Figure 9: Additional first-order search interpretations on random image pairs from the Pascal VOC dataset

## A.7 ADDITIONAL RESULTS FOR STANFORD ONLINE PRODUCTS

Table 2: Comparison of performance of first-order search interpretation methods across different visual search systems on the Stanford Online Product dataset. Methods introduced in this work are highlighted in pink. *Though SAM generalizes Zhu et al. (2019) we refer to it as a baseline. For additional details see Section 6

| Metric | Model | SBSM | PSHAP | LIME | KSHAP | VESM | GCAM | SAM* | IG SAM | KSAM |
|---|---|---|---|---|---|---|---|---|---|---|
| | | | Model Agnostic | | | | Architecture Dependent | | | |
| Faith. | DN121 | 0.18 | **0.23** | 0.20 | 0.22 | 0.09 | 0.13 | 0.12 | **0.18** | **0.18** |
| | MoCoV2 | 0.24 | **0.30** | 0.27 | 0.18 | 0.14 | 0.2 | 0.21 | **0.24** | **0.24** |
| | RN50 | 0.11 | **0.14** | 0.12 | 0.13 | 0.03 | 0.07 | 0.07 | **0.10** | **0.10** |
| | VGG11 | 0.15 | **0.16** | 0.14 | 0.15 | 0.04 | 0.08 | 0.09 | **0.12** | **0.12** |
| Ineff. | DN121 | - | **0.00** | 0.24 | **0.00** | - | 11.2 | 0.54 | 0.02 | **0.00** |
| | MoCoV2 | - | **0.00** | 0.17 | **0.00** | - | 0.34 | 0.57 | 0.02 | **0.00** |
| | RN50 | - | **0.00** | 0.21 | **0.00** | - | 13.6 | 0.39 | 0.02 | **0.00** |
| | VGG11 | - | **0.00** | 0.24 | **0.00** | - | 4.13 | 0.47 | 0.04 | **0.00** |

## A.8 ADDITIONAL RESULTS FOR CALTECH-UCSD BIRDS 200 (CUB) DATASET

Table 3: Comparison of performance of first-order search interpretation methods across different visual search systems on the CUB dataset. Methods introduced in this work are highlighted in pink. *Though SAM generalizes Zhu et al. (2019) we refer to it as a baseline. RN50-ML refers to a ResNet50 architecture trained for metric learning on the CUB dataset with the margin loss Roth et al. (2020). For additional details see Section 6

| Metric | Model | SBSM | PSHAP | LIME | KSHAP | VESM | GCAM | SAM* | IG SAM | KSAM |
|---|---|---|---|---|---|---|---|---|---|---|
| | | | Model Agnostic | | | | Architecture Dependant | | | |
| Faith. | DN121 | 0.25 | **0.38** | 0.31 | 0.34 | 0.15 | 0.10 | 0.12 | **0.30** | **0.30** |
| | RN50-ML | 0.39 | **0.49** | 0.47 | **0.49** | 0.04 | 0.14 | 0.17 | **0.41** | **0.41** |
| | MoCoV2 | 0.32 | **0.47** | 0.39 | 0.41 | 0.26 | 0.26 | 0.26 | **0.34** | **0.34** |
| | RN50 | 0.14 | **0.21** | 0.18 | 0.18 | 0.05 | 0.07 | 0.07 | **0.14** | **0.14** |
| | VGG11 | 0.23 | **0.31** | 0.26 | 0.27 | 0.11 | 0.15 | 0.16 | **0.23** | 0.22 |
| Ineff. | DN121 | - | **0.00** | 0.17 | **0.00** | - | 16.0 | 0.58 | 0.02 | **0.00** |
| | RN50-ML | - | **0.00** | 0.13 | **0.00** | - | 5.23 | 0.48 | 0.03 | **0.00** |
| | MoCoV2 | - | **0.00** | 0.19 | **0.00** | - | 0.44 | 0.60 | 0.03 | **0.00** |
| | RN50 | - | **0.00** | 0.15 | **0.00** | - | 15.5 | 0.43 | 0.02 | **0.00** |
| | VGG11 | - | **0.00** | 0.17 | **0.00** | - | 4.25 | 0.54 | 0.05 | **0.00** |

## A.9   ADDITIONAL RESULTS FOR MS COCO

Table 4: Comparison of performance of first and second-order search interpretation methods across different visual search systems on the MSCOCO dataset. Methods introduced in this work are highlighted in pink. *Though SAM generalizes Zhu et al. (2019) we refer to it as a baseline. For additional details see Section 6

| Metric | Order | Model | SBSM | PSHAP | LIME | KSHAP | VESM | GCAM | SAM* | IG SAM | KSAM |
|---|---|---|---|---|---|---|---|---|---|---|---|
| | | | | Model Agnostic | | | | Architecture Dependent | | | |
| Faithfulness | First | DN121 | 0.18 | **0.24** | 0.22 | 0.22 | 0.10 | 0.12 | 0.11 | 0.14 | **0.17** |
| | | MoCoV2 | 0.25 | **0.37** | 0.33 | 0.35 | 0.15 | 0.23 | 0.24 | 0.24 | **0.28** |
| | | RN50 | 0.10 | **0.14** | 0.12 | 0.12 | 0.04 | 0.07 | 0.07 | 0.07 | **0.09** |
| | | VGG11 | 0.14 | **0.15** | 0.14 | 0.14 | 0.05 | 0.09 | 0.1 | 0.10 | **0.12** |
| | Second | DN121 | 0.49 | - | **0.57** | 0.57 | - | - | 0.5 | **0.51** | 0.49 |
| | | MoCoV2 | 0.73 | - | **0.79** | 0.79 | - | - | 0.77 | 0.77 | **0.78** |
| | | RN50 | 0.73 | - | **0.78** | 0.78 | - | - | **0.75** | 0.75 | 0.73 |
| | | VGG11 | 0.67 | - | **0.73** | 0.73 | - | - | 0.71 | 0.71 | **0.72** |
| Inefficiency | First | DN121 | - | **0.00** | 0.22 | **0.00** | - | 12.3 | 0.6 | 0.02 | **0.00** |
| | | MoCoV2 | - | **0.00** | 0.11 | **0.00** | - | 0.46 | 0.66 | 0.02 | **0.00** |
| | | RN50 | - | **0.00** | 0.22 | **0.00** | - | 15.8 | 0.47 | 0.02 | **0.00** |
| | | VGG11 | - | **0.00** | 0.31 | **0.00** | - | 3.47 | 0.59 | 0.04 | **0.00** |
| | Second | DN121 | - | - | 0.15 | **0.01** | - | - | 0.20 | 0.01 | **0.00** |
| | | MoCoV2 | - | - | 0.10 | **0.01** | - | - | 0.09 | 0.02 | **0.00** |
| | | RN50 | - | - | 0.07 | **0.01** | - | - | 0.07 | 0.02 | **0.00** |
| | | VGG11 | - | - | 0.11 | **0.01** | - | - | 0.19 | 0.04 | **0.00** |
| mIoU | Second | DN121 | 0.50 | - | **0.62** | 0.61 | - | - | 0.62 | **0.63** | 0.52 |
| | | MoCoV2 | 0.52 | - | **0.62** | 0.61 | - | - | 0.64 | **0.66** | 0.60 |
| | | RN50 | 0.50 | - | **0.62** | 0.61 | - | - | 0.63 | **0.65** | 0.48 |
| | | VGG11 | 0.50 | - | **0.62** | 0.61 | - | - | 0.66 | **0.67** | 0.60 |

## A.10 ADDITIONAL KERNEL CONVERGENCE RESULTS

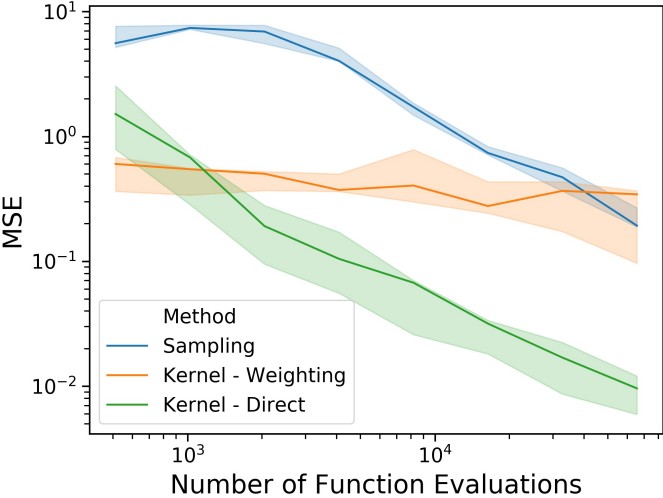

Figure 10: Kernel convergence for random functions generated by randomly choosing coefficients. Results generally mirror those for randomly initialized deep networks

## A.11 GENERALIZING SBSM TO JOINT SEARCH ENGINE INTERPRETABILITY

Before generalizing SBSM (Dong et al.) to joint interpretability we will review the original implementation for marginal interpretability. SBSM uses a sliding square mask and multiple evaluations of the search engine to determine which regions of the image are important for similarity. More formally, let $q$, and $r$ represent the pixels of the query image and retrieved image. Let $M_{ij}^s(q)$ represent the result of replacing a square of pixels of size $s \times s$ centered at pixel $(i, j)$ with a "background value" which in our case is black. SBSM "slides" this mask across the query image and compares the similarity between the masked query and retrieved image. These masked similarity values are compares to the baseline similarity value and stored in a weight matrix, $w$:

$$w_{ij} = \min \left[ d \left( M_{ij}^s(q), r \right) - d \left( q, r \right), 0 \right] \tag{10}$$

Intuitively speaking, the weights $w_{ij}$ represent the impact of masking a square centered at $(i, j)$. For areas that are critical to the similarity, this will result in $w_{ij} > 0$. Finally, an attention mask on the query image is formed by a weighted average of the masks used to censor the images. For square masks, this can be achieved efficiently using a deconvolution with a kernel of ones of size $s \times s$ on the weight matrix $w$. We also note that instead of evaluating the (expensive) distance computation $d$ for every pixel $(i, j)$, one can also sample pixels to censor. We use this approach in our joint generalization.

To generalize SBSM we use a pair of masks, one for the query image and one for the retrieved image respectively. We sample mask locations and calculate weights designed to capture the intuition that censoring corresponding areas cause similarity to increase as opposed to decrease. More specifically we use the following weighting scheme:

$$w_{ij}^{hw} = \min \left[ d \left( q, r \right) - d \left( M_{ij}^s(q), M_{hq}^s(r) \right), 0 \right] \tag{11}$$

Because evaluating the similarity function for every $(i, j, h, w)$ combination is prohibitively expensive, we instead sample masked images for our computation. To project attention from a query pixel, we query for all masks that overlap with the selected query pixel, and then average their corresponding retrieved masks according to the weights calculated in Equation 11.

## A.12 PROOF OF PROPOSITION 4.1

Let $\mathcal{X} = \mathcal{Y} = \mathbb{R}^{CHW}$ and represent the space of deep network features where $C, H, W$ represent a channel, height, and width of the feature maps respectively. Let the function $d := \sum_c GAP(x)_c GAP(y)_c$. Let the grand coalition, $N = [0, H] \times [0, W]$, index into the spatial coordinates of the image feature map $y$. Let the function $mask(y, S)$ act on a feature map $y$ by replacing the features at locations $S$ with a background signal $b$. For notational convenience let $\psi_i(v) := \phi_v(i)$ represent the Shapley value the $i^{th}$ player under the value function $v$. We begin by expressing the left-hand side of the proposition:

$$\psi_{hw}(v) = \psi_{hw}\left(\sum_c GAP(x)_c GAP(mask(y, S))_c\right)$$

$$= \psi_{hw}\left(\frac{1}{HW}\sum_c GAP(x)_c \sum_{h'w'} mask(y, S)_{ch'w'}\right)$$

$$= \frac{1}{HW}\sum_{h'w'} \psi_{hw}\left(\sum_c GAP(x)_c mask(y, S)_{ch'w'}\right) \qquad \text{(Linearity)}$$

$$= \frac{1}{HW}\psi_{hw}\left(\sum_c GAP(x)_c mask(y, S)_{chw}\right) \qquad \text{(Dummy)}$$

$$= \frac{1}{HW}\sum_c GAP(x)_c(y_{chw} - b_{chw}) \qquad \text{(Efficiency)}$$

## A.13 PROOF OF PROPOSITION 5.1

Let the spaces $\mathcal{X}$, $\mathcal{Y}$ and function $d$ be as in Proposition 4.1. As a reminder the function $d$ represents the un-normalized GAP similarity function. Let the grand coalition, $N$, index into the spatial coordinates of both the query image features $x \in R^{CHW}$ and retrieved image features $y \in R^{CHW}$. Let the function $mask(y, S)$ act on a feature map $y$ by replacing the corresponding features with a background feature map $a$ for query features and $b$ for retrieved features. We can represent the set of players, $N$, as a set of ordered pairs of coordinates with additional information about which tensor, the query (0) or retrieved (1) features, they represent:

$$N = ([1, H] \times [1, W] \times \{0\}) \cup ([1, H] \times [1, W] \times \{1\}) \qquad (12)$$

In the subsequent proof we omit these 0, 1 tags as it is clear from our notation which side, query or retrieved, the index refers to based on the index $h, w$ for the query and $i, j$ for the retrieved image. We first consider the zero background value function, $v(S \subset N)$, defined by censoring the spatially varying features prior to global average pooling and comparing their inner product:

$$v(S) = \left(\frac{1}{HW}\sum_{h,w} \tilde{x}_{chw}\right) \cdot \left(\frac{1}{HW}\sum_{i,j} \tilde{y}_{cij}\right)$$

where

$$\tilde{x}_{chw} = \begin{cases} x_{chw} & (h, w) \in S \\ 0 & o.w. \end{cases}$$

and likewise, for $y_{cij}$. When $S$ contains all $i, j, h, w$ this represents the similarity judgement from the GAP network architecture. We seek the Shapley-Taylor index for a pair of image locations $S = \{(h, w), (i, j)\}$. For notational convenience let $\psi_S^k(v) := \phi_v^k(S)$ represent the $k-$order interaction

effects for the subset $S$ and the value function $v$.

$$
\begin{aligned}
\psi_S^k(v) &= \psi_S^k\left(\left(\frac{1}{HW}\sum_{h',w'}\tilde{x}_{ch'w'}\right)\cdot\left(\frac{1}{HW}\sum_{i',j'}\tilde{y}_{ci'j'}\right)\right)\\
&= \psi_S^k\left(\frac{1}{H^2W^2}\sum_c\sum_{h',w'}\sum_{i',j'}\tilde{x}_{ch'w'}\tilde{y}_{ci'j'}\right)\\
&= \psi_S^k\left(\frac{1}{H^2W^2}\sum_{h',w'}\sum_{i',j'}\sum_c\tilde{x}_{ch'w'}\tilde{y}_{ci'j'}\right)\\
&= \sum_{h',w'}\sum_{i',j'}\psi_S^k\left(\sum_c\frac{1}{H^2W^2}\tilde{x}_{ch'w'}\tilde{y}_{ci'j'}\right) && \text{(Linearity)}\\
&= \psi_S^k\left(\sum_c\frac{1}{H^2W^2}\tilde{x}_{chw}\tilde{y}_{cij}\right) && \text{(Dummy)}\\
&= \psi_{S'}^k(v_{hwij}) && \text{(Renaming)}
\end{aligned}
$$

Where the renaming of the last step was because we can now consider a simplified value function with just the non-dummy players as $v_{hwij}(S') := \sum_c \tilde{x}_{chw}\tilde{y}_{cij}$. Where $S'$ represents a subset of the non-dummy players: $N' = \{(h,w),(i,j)\}$. We can now explicitly calculate the index:

$$
\begin{aligned}
\psi_{S'}^2(v) &= \frac{2}{n}\sum_{T\subseteq N'\backslash S'}\delta_{S'}v_{hwij}(T)\frac{1}{\binom{n-1}{t}}\\
&= \delta_{S'}v_{hwij}(\emptyset)\\
&= \frac{1}{H^2W^2}\sum_c x_{chw}y_{cij}
\end{aligned}
$$

By following the same set of reasoning, we can introduce nonzero background values $a_{chw}$ and $b_{cij}$ to yield the following:

$$
\psi_{hw,ij}^2(v) = \frac{1}{H^2W^2}\sum_c x_{chw}y_{cij} - x_{chw}b_{cij} - a_{chw}y_{cij} + a_{chw}b_{cij} \tag{13}
$$

## A.14 Proof that Shapley Taylor is Proportional to Integrated Hessians for GAP architecture

As in Proposition 4.2 we consider the soft masking or "multilinear extension" of our second-order value function $v_2$:

$$
v_2(S): [0,1]^N \to \mathbb{R} := d(mask(x,S),mask(y,S)) \tag{14}
$$

let $hw$, and $ij$ be members of the grand coalition $N$ such that $hw \neq ij$. We begin our proof with the expression for the off-diagonal terms of the Integrated Hessian.

$$
\Gamma_{hw,ij}(S) := \int_{\alpha=0}^1\int_{\beta=0}^1 \alpha\beta\frac{\partial^2 v_2(\alpha\beta S)}{\partial T_{hw}\partial T_{ij}} \tag{15}
$$

Where $\frac{\partial}{\partial T_h w}$ represents the $hw$ component of the partial derivative with respect to $\alpha\beta S$, not to be confused with the partial derivative of just S. Like in our proof of Proposition 4.2, because our function is defined on the interval $[0,1]^N$ many of the terms mentioned in Janizek et al. (2020) drop out and instead are captured in the Hessian of the function with repsect to the soft mask. We now

expand the definition of $v_2(\alpha\beta S)$:

$$v_2(\alpha\beta S) = d(mask(x, \alpha\beta S), mask(y, \alpha\beta S))$$

$$= \frac{1}{H^2W^2} \sum_c \left( \sum_{h,w} a_{chw} + \alpha\beta S_{hw}(x_{chw} - a_{chw}) \right) \left( \sum_{i,j} b_{cij} + \alpha\beta S_{ij}(y_{cij} - b_{cij}) \right)$$

From this function we can read off the appropriate term of the hessian with respect to the mask at location $(h, w)$ and location $(i, j)$

$$\frac{\partial^2 v_2(\alpha\beta S)}{\partial T_{hw} \partial T_{ij}} = \frac{1}{H^2W^2} \sum_c x_{chw}y_{cij} - x_{chw}b_{cij} - a_{chw}y_{cij} + a_{chw}b_{cij}$$

$$= \psi^2_{hw,ij}(v)$$

We can now pull this outside the integral to yeild:

$$\Gamma_{hw,ij}(v_2) = \int_{\alpha=0}^{1} \int_{\beta=0}^{1} \alpha\beta \frac{\partial^2 v_2(\alpha\beta S)}{\partial T_{hw} \partial T_{ij}}$$

$$= \psi^2_{hw,ij}(v) \int_{\alpha=0}^{1} \int_{\beta=0}^{1} \alpha\beta$$

$$= \frac{1}{4} \psi^2_{hw,ij}(v)$$

Which proves that the Shapley-Taylor index and second order Aumann-Shapley values are proportional for the GAP architecture.

## A.15 EXPLAINING DISSIMILARITY

In addition to explaining the similarity between two images, our methods naturally explain image dissimilarity. In particular, regions with a negative Shapely values (Blue regions in Figure 11) contribute negatively to the similarity between the two images. These coefficients can be helpful when trying to understand why an algorithm does not group two images together.

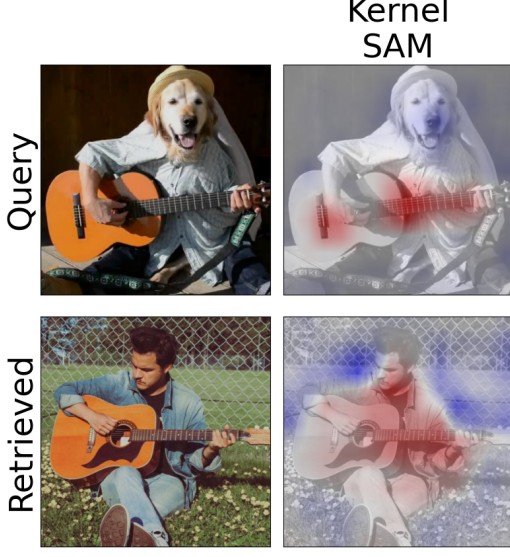

Figure 11: Explanation of why two images are similar (Red) and dissimilar (Blue). Blue regions highlight major differences between the images such as the dog playing the guitar, and the chain-link fence in the retrieved image.

### A.16 On the "Axiomatic" terminology

The term "axiomatic" can mean different things to different readers. When this work refers to "axiomatic" methods we refer to methods that approximate the uniquely specified explanation values dictated by the axioms of fair-credit assignment. In the first-order case, these explanations are the Shapey Values and satisfy the axioms of linearity, efficiency, dummy, and symmetry. In the higher-order case these fair credit assignments are the Shapley-Taylor Indices and satisfy analogous axioms Sundararajan et al. (2020). We note that our methods *converge* to the true Shapley and Shapley-Taylor indices and thus the deviations that arise as part of convergence induce corresponding deviations from the axioms of fair credit assignment. Nevertheless, we find that these deviations become negligible as our methods converge to the true Shapley and Shapley-Taylor values. This starkly contrasts the behavior of methods that do not converge to values that satisfy the axioms of fair credit assignment such as GradCAM as shown in Figure 8.

