# OpenReview forum: "Axiomatic Explanations for Visual Search, Retrieval, and Similarity Learning"
_ICLR.cc/2022/Conference — ICLR 2022 Poster_

### Official Review · Reviewer_erSL · 2021-11-01

**Correctness:** 3
**Technical Novelty And Significance:** 4
**Empirical Novelty And Significance:** 3
**Recommendation:** 6
**Confidence:** 4

**Main Review:**

Strengths:
+ This work builds up a formal explanation framework for retrieval/metric learning, which can benefit future works in this field. It also generalizes other explanation methods, like CAM, GradCAM, VEDML (generalized as SAM in this paper).
+ The idea of using Shapley value to explain retrieval model is very interesting.
+ The definitions and propositions are reasonable, I did not find any errors there.
+ The experiments provide a good comparison between all existing and generalized methods on faithfulness, quality as well as the efficiency.
+ The writing is easy to follow and well organized.

Concerns:
- The claim of "Axiomatic" is very strong. Shapley value may be a fair metric to measure the contribution of each feature, but the value in this work is estimated with approximator. Is the estimated value guaranteed to be "Axiomatic" here?
- There are many principles to follow for explanation, and Shapley value is not the only reasonable one. For example, perturbation [1], and Deep Taylor Decomposition principle [2]. Why is Shapley value called "Axiomatic" while others not?
1. Fong, Ruth C., and Andrea Vedaldi. "Interpretable explanations of black boxes by meaningful perturbation." Proceedings of the IEEE international conference on computer vision. 2017.
2. Chefer, Hila, Shir Gur, and Lior Wolf. "Transformer interpretability beyond attention visualization." Proceedings of the IEEE/CVF Conference on Computer Vision and Pattern Recognition. 2021.

- The performance is not improved. In table 1, it seems that the proposed kernel-SAM only improves efficiency over SAM, the quality and faithfulness are both on par with second-order SAM. Since the experiment is conduct with GAP architecture, the SAM is equivalent to VEDML (in related work), it this correct? If yes, then the performance is not improved over VEDML. In scenarios where efficiency is not important, we still don't know how to get a better explanation.
- The proposed method is claimed to have advantage over other methods when the architecture is not based on GAP. But the experiment is only conducted with GAP-based architecture. Why not try other architectures where the proposed method could have better performance?
- The explanation framework is for retrieval/metric learning/similarity models, while this paper only conducts experiments on object detection datasets (VOC, COCO). It would be more convincing if results on widely used retrieval/metric learning datasets are provided, e.g. CUB, Deepfashion.
- The authors use Moco, a contrastive learning model for self-supervised learning. It uses augmented images as positive and all the other images as negative, which is different from normal retrieval model. The results on retrieval model may be different. Why not train with metric learning setting, i.e. using the same class as positive while other classes as negative (contrastive loss, triplet loss, proxy).
-  Proposition 4.2 raises a limitation about GradCAM. It seems that VEDML also mentioned the limitation of GradCAM about the GAP operation. Although their definition is not as formal as this work, it somehow aligns with the observation of this work.


**Summary Of The Paper:**

This paper proposes a explanation framework for retrieval/similarity/metric learning models. It generalizes the form of a couple of previous works, e.g. Zhou et al., 2016; Selvaraju et al., 2017; Zhu et al., 2019; Ribeiro et al., 2016. The proposed method relies on estimation of Shapley value, the authors then propose a kernel-based approximator to make it more computationally efficient. Experiments are conducted on PascalVOC and MSCoCo dataset to show the faithfulness and quality of explanations.

**Summary Of The Review:**

Overall, I think the idea is very interesting and the proposed explanation method is good contribution. But the experiment is not strong due to the concerns, and the evaluation could be improved in the future. My rating is borderline leaning to accept.

---

> ### Author Response · Authors · 2021-11-19
> **Clarifying the term Axiomatic, Additional Experiments, Clarifications on the GAP Architecture, and other improvements.**
>
> Thank you for your thoughtful comments on our work. We have updated our manuscript and marked major changes with yellow highlights.
>
> ## On the term “Axiomatic”:
>
> We wanted to clarify exactly what we mean by the term “Axiomatic”. In particular, we refer to our methods as Axiomatic because they converge to the unique explanation values guaranteed by the axioms of fair credit assignment, a well-established unifying framework for fiar model explanations [1]. This nomenclature aligns with other prior works in the literature such as [1] which introduces Shapley Value approximators for classifiers.  To understand how well our approaches approximate these uniquely specified explanations, we measure their alignment with the axioms using metrics such as Faithfulness and Inefficiency. In most cases, we observe that axioms like efficiency are generally satisfied by our approaches and violated by other approaches in the literature. Furthermore, our kernel convergence figures demonstrate that our approaches rapidly converge to the true values.
>
> We appreciate you highlighting the work of Fong et. al. and Chefer et. al. and have added these to our related work section. These methods introduce particular lenses for deriving explanation methods they do not tie them to the axioms of fair credit assignment. If the aforementioned works do not implicitly approximate the Shapley Value, then they will violate one of the axioms of fair credit assignment (Efficiency, Linearity, Dummy, and Symmetry) in some case. We note that it is valid to question the axioms of fair credit assignment and we do not wish to claim that this is the only way to view model explanations, however, we are interested in these axioms because of their well-established unifying role in the literature [1].
>
>
>
> [1] Lundberg, Scott M., and Su-In Lee. "A unified approach to interpreting model predictions." Proceedings of the 31st international conference on neural information processing systems. 2017.
>
>  ##  Additional Experiments on CU Birds and Stanford Online Products:
>
> We have added additional experiments on the CU Birds and Stanford Online Products datasets as requested. Though these datasets do not contain the necessary information to evaluate second-order explanations we have compared the performance of our first-order methods. In general, we find that explanations based on our fair credit assignments better capture the relevant areas of the image and yield a larger change in the model’s score when blurred. The methods also better satisfy the Efficiency axiom of fair credit assignment.
>
>  ##  Clarifying questions about the GAP Architecture
> The reviewer mentioned that our experiments use the GAP architecture but we want to clarify that our experiments use cosine similarity. This architecture is not technically the GAP architecture as described in the theorems because cosine similarity adds a nonlinear normalization to the inner product. In this setting, VEDML/SAM diverge from the axioms of fair credit assignment which is why their inefficiency rises. The authors of VEDML mention that GradCAM and VEDML differ in the cosine similarity case but they do not show that both violate the axioms of fair credit assignment in the cosine nonlinearity case. Furthermore, they do not mention that GradCAM’s spatial averaging of gradients makes it violate the dummy axiom of fair credit assignment.
>
> ## On the performance of axiomatic methods
> Though the gains of our methods for second-order faithfulness are slight, we note that in the first-order case these methods are dramatically more faithful to the underlying model and do not violate the efficiency axiom. We also note that for black-box model explainability our second-order methods outperform the baseline SBSM in all metrics.
>
> ## On the choice of models
> Our experimental setting features some of the most commonly used image features in the computer vision and self-supervised learning community. We stress that whether we use these models or other similarity models from the literature, the theoretical guarantees of our approaches still hold, and other methods will still fail the axioms of fair credit assignment in measurable ways. We argue that the specific choice of image similarity model is not as important as the fact that the experimental conditions are fair.
>
>  ## Explaining Dissimilarity:
> We have added Section A.15 in the appendix to highlight how our methods can explain image dissimilarity. In particular, areas with a negative Shapley value can be interpreted as lowering the overall similarity of the two images. Thus these negative values are the regions that contribute most to the “dissimilarity” of the two images.
>
> Thank you again for your thoughtful feedback and we kindly ask that you consider raising your score if you found these comments helpful.

---

> > ### Comment · Reviewer_erSL · 2021-11-19
> > **Response**
> >
> > I appreciate the rebuttal, here is my response.
> >
> > “Axiomatic”- I think this is not well addressed. “Axiomatic” means something is always true, which is very strong. Assume that the fair credit assignment theory is axiomatic, then the estimator has 99% accuracy on estimating the Shapley Value, there is 1% chance that the estimator will fail. Can we still say this is axiomatic?
> >
> > Additional Experiments - These new results are good.
> >
> > GAP - I think the authors misunderstand my point. My comment says that the paper claims that the proposed method is not dependent on architecture, which is an advantage over other methods. Experiments should be conducted to show this. Only one architecture cannot validate this. In addition, I don’t think using cosine similarity is anything special, as it is commonly used for retrieval method as well as retrieval explanation methods. Why is cosine similarity a non-linear normalization to inner product? The difference is just the L2 norm which is a linear scalar.
> >
> > Performance - I think the advantage of the proposed method is on faithfulness, not performance. Could be a limitation, not a concern.
> >
> > Choice of models - I think the authors did not get my point. This paper proposes general framework for all retrieval/metric learning/similarity models. If only one model can be evaluated in this paper, then it should be a general retrieval/metric learning model. The experiments on self-supervised model are good, but they are just using contrastive learning as a pre-text task. They do not care about the retrieval performance, but only care about the generalization performance on downstream tasks. For general retrieval, what we care is retrieval performance itself. That is why these models are different. Readers would be interested to see how well a retrieval model trained on retrieval datasets can be explained with the proposed method. Hope my point is clear here.
> >
> > Explaining Dissimilarity – Good qualitative result. Very interesting.

---

> > > ### Author Response · Authors · 2021-11-19
> > > **Additional Section and more clarification**
> > >
> > > Thanks for your fast and thoughtful response. We have made a few more modifications based on your clarifications
> > >
> > > ## Additional Section to Clarify the meaning of “Axiomatic”
> > > Yes you are correct and we have added an additional section in the appendix to try to clarify this. Thank you for clarifying your thoughts so that we could better address them. To be precise, these formulations converge to quantities that satisfy the axioms, but if they are not properly converged then they will only satisfy them approximately. Because of this we have tried to measure the violation using the inefficiency metric which is very low or zero for our methods. Nevertheless we hope that our section in the appendix on GradCAM violating the Dummy axiom will give a sense of just how much current approaches violate these axioms and how our approach does not suffer the same large violations. We hope that this additional section will provide the appropriate nuance to our terminology.
> > >
> > > ## On the nonlinearity of Cosine similarity
> > > The reviewer is correct in their judgement that cosine similarity is a commonly used distance metric for retrieval methods. That is exactly why we focus on it in this work. In some sense it is the most popular “nonlinearity” added on top of a retrieval architecture. This nonlinearity has the effect of making SAM and fair credit assignment diverge, thus is why we are interested in investigating it.
> > > We wanted to demonstrate that cosine normalization is indeed nonlinear, despite the intuition of it as a scalar multiplication. The key detail is that the scalar multiplying factor to normalize is nonlinear. More precisely, consider the equation for the normalizing factor for a particular vector $x \in \mathbb{R}^n$:
> > > $$f(x) = \frac{1}{\sqrt{ \sum_i x_i^2}}$$
> > >
> > > This function nonlinear due to the presence of a square root and a reciprocal. One can check that:
> > >
> > > $$f(ax + by) \neq af(x) + bf(y)$$
> > >
> > > A quick example shown below for the vectors x = [1,0] y=[0,1], a = 1, b=1
> > >
> > > $$f(ax +by) = f([1,1]) = [\frac{1}{sqrt(2)}, \frac{1}{sqrt(2)}]$$
> > >
> > > Whereas:
> > >
> > > $$af(x) +bf(y) = [1,1]$$
> > >
> > > In general which demonstrates this explicitly. If the vectors in the normalizing term were constant then yes, the operation would be a trivial rescaling of the vectors but this is not the case for the standard cosine normalized features. We hope this clarifies this somewhat tricky mathematical point and demonstrates that we have evaluated on a “nonlinear” architecture.
> > >
> > > ## On the choice of model
> > > We appreciate your clarification of what you would like to see in the work. We have begin the process of adding several metric learning methods trained on the CUB and Stanford Online products dataset. Though we might not be able to have these results by the 22nd we give you our word that the camera ready copy will have these models and hope that you could still consider raising your score to help us share this work with the broader community.
> > >
> > > Thank you again for your discussions and we hope that these additions can make you more confident in the work.

---

> > > > ### Comment · Reviewer_erSL · 2021-11-29
> > > > **Response**
> > > >
> > > > I have read the reply from authors, and would like to keep my rating.
> > > > - The cosine similarity is not special, my point is that experiment is needed to support the claim that the method is architecture-independent. Current version only has one architecture, it would be better to show results on other architectures.
> > > > - I appreciate the author's hard work. Hopefully the results can be included in the final camera ready version.

---

### Official Review · Reviewer_T6UD · 2021-11-07

**Correctness:** 3
**Technical Novelty And Significance:** 3
**Empirical Novelty And Significance:** 2
**Recommendation:** 6
**Confidence:** 5

**Main Review:**

***Strengths***

Providing a general framework grounded in some theory for adapting existing work to a new task ensures that their approach can likely continue to provide value over time (since as new classification explanation methods are developed they could adapt those too).  The paper is also pretty well written and easy to understand, and addresses a problem with wide applications and relatively little work, making the added discussion quite valuable on its own.  The experiments themselves do suggest their approach has merit, although lacking in several aspects in this version.

***Weaknesses***

1. The authors never showed their explanation method was human-interpretable or could be used in any downstream methods that leverage explanations.  This is a critical flaw in their paper that is enough to recommend rejection on its own.  The authors try to avoid this by saying people introduce biases when considering explanations so is out of scope, but I am going to flatly reject this line of reasoning.  If it is an explanation, then one factor should be human interpretability, since it is one of its primary applications.  I would be willing to accept an alternative, such as demonstrating that this kind of approach would be useful for methods that utilize explanations during training (the authors referenced a few, so I will omit citing examples here), but these experiments have yet to be conducted as well.

2. Note the authors failed to discuss and cite another paper that introduced a general framework for generalizing prior work for explaining similarity models:

Bryan A. Plummer, Mariya I. Vasileva, Vitali Petsiuk, Kate Saenko, David Forsyth. Why do These Match? Explaining the Behavior of Image Similarity Models. ECCV, 2020.

The authors should also discuss and compare to this paper.

3.  Following up on the paper in the second weakness, the authors of that paper found saliency-based methods (such as this paper) is not actually human interpretable for the image similarity setting, and could actually lead to more confusion when used in isolation.  While this could simply be due to the biases the authors argued as a reason not to do human studies, this should be confirmed as it suggests that the approach used by this paper may lead to similar results.  Some discussion by authors on this point is warranted.

4. The authors seem to mostly consider the case for explaining why two images are similar, but not what causes dissimilarity.  For example, some key image region A could be neglected by image B, producing a poor similarity score.  The authors should include a discussion and experiments that handle this case.

5. The authors perform experiments on two object detection benchmarks, which makes their claim of an image similarity explanation model a bit suspect.  They should also include experiments over standard deep metric learning benchmarks at the very least (e.g., Deepfashion, Stanford Online Products, CUB, etc), although I have a strong preference for medical images since this case was used repeatedly in the motivation for this paper.

***Post Rebuttal***
For the most part I was satisfied with the rebuttal.  The approach has not been demonstrated that it is human-interpretable, meaning that one of the major applications of explainable methods are not verified.  In addition, the authors did not compare to the prior work that I suggested.  While the authors argue that their explanations are better, and I don't necessarily doubt it, but demonstrating it empirically would make a stronger paper.  I would encourage the authors to at the very least discuss and (hopefully) compare to the paper I cited as it a published paper that addresses the same topic as this paper and currently is (still) uncited.


**Summary Of The Paper:**

This paper proposes an approach for explaining visual similarity models.  They provide a general framework that enables them to adapt existing explanation methods to the similarity learning task, and then discuss how they apply their framework to generalize multiple methods.  The authors evaluate their work on PASVAL VOC and COCO, where they use automatic metrics to argue that their approach accurately captures model performance.

**Summary Of The Review:**

My initial recommendation is to reject this paper.  The authors have not demonstrated that their approach provides useful explanations either via a human evaluation or for some downstream task.  They are missing a reference to at least one work addressing the same task that has claims that are problematic for this paper, and the experiments were not conducted on datasets that were designed for the motivating applications as well, making it unclear if the approach will generalize to the target setting.  The authors need address at least the first two weaknesses before I would consider recommending acceptance, although the 5th weakness is also high on my priority list for changes.

---

> ### Author Response · Authors · 2021-11-19
> **Highlighting Downstream Tasks, Explaining Dissimilarity, Additional Experiments**
>
> Rebuttal
>
> Thank you for your thoughtful comments on our work. We have updated our manuscript and marked major changes with yellow highlights.
>
> ## 1) On Human Interpretability and usefulness for downstream tasks:
> We appreciate your suggestion to perform human evaluations and do not disagree with your sentiment that Human evaluations would help this work. However, given the limited time to modify this work and the complexities associated with human experiments we chose instead to try and enhance our demonstration of how these approaches help downstream tasks.
> In particular, we wanted to draw your attention to the existing evaluation of downstream task performance in table 1. The mIoU metric measures how well second-order interpretability can be used for semantic segmentation label propagation. These positive results show that the explanations align with human annotations on the Pascal VOC and MSCOCO datasets, and we have tried to expand on that in our revision. We have also begun work on a second application experiment related to segmentation, but finishing it during the review period is unlikely.
>
> We should mention that we have also written follow-up work to this paper that embeds these second-order explanations within a self-supervised visual transformer architecture to significantly surpass the current state of the art (+14 mIoU) in unsupervised semantic segmentation on both the MSCOCO and Cityscapes datasets. This combined work was way too large to reasonably fit within a single paper so we decided to split the theoretical analysis (this work) from the downstream applications (Our follow-up work). We did not include the citation in this work to preserve anonymity, but we will include a pointer to this application paper in the final work that we hope helps readers gauge usefulness in specific downstream applications.
>
> We hope that future work can investigate human evaluation and take these nuances into consideration. Nevertheless, we hope our previous efforts to evaluate our approaches downstream tasks can help move the paper in a direction you are comfortable with.
>
>  ## 2) Regarding Plummer et al.:
>
> We appreciate you identifying another first-order baseline and are in the process of onboarding this to our experimental setting and will update the table once completed. We stress that though this is an additional baseline, the work is not directly comparable to the richer second-order similarity explanations put forth by our work, and so we believe both works present independent value to the community.
>
> ## 3) Regarding the human interpretability of saliency-based methods
> We wanted to clarify here that we do not assert that salience maps are the only solution to the question of model explainability. In contrast, we view these visualizations as a part of a broader strategy to understand a model. Saliency maps appear throughout the literature and are commonly used because it is reasonable to ask “where is my model looking”.  By correcting the computation of these quantities to better align with axiomatic fairness, we hope to improve these visualizations (and their second-order extensions) for the community. However, we do not claim that this is the only approach to explaining models and welcome other research in methods such as verbal explanations, direct counterfactuals, etc.
>
> ## 4) Explaining Dissimilarity
>
> We have added Section A.15 in the appendix to highlight how our methods already explain image dissimilarity. In particular, areas with a negative Shapley value can be interpreted as lowering the overall similarity of the two images. Thus these negative values are the regions that contribute most to the “dissimilarity” of the two images.
>
>  ##  5) Additional Experiments on CU Birds and Stanford Online Products:
>
> We have added additional experiments on the CU Birds and Stanford Online Products datasets as requested. Though these datasets do not contain the necessary information to evaluate second-order explanations we have compared the performance of our first-order methods. In general, we find that explanations based on our fair credit assignments better capture the relevant areas of the image and yield a larger change in the model’s score when blurred. The methods also better satisfy the Efficiency axiom of fair credit assignment.
>
>
> Thank you again for your thoughtful feedback, and for being clear about what you thought needed to be addressed to be suitable for acceptance.

---

> ### Author Response · Authors · 2021-11-29
> **Following Up**
>
> Thank you for your helpful comments, as today is the last day of the review please let us know if there is anything else we can address at this time. We appreciate your consideration

---

### Official Review · Reviewer_EVSE · 2021-11-09

**Correctness:** 4
**Technical Novelty And Significance:** 3
**Empirical Novelty And Significance:** 3
**Recommendation:** 6
**Confidence:** 2

**Main Review:**

This work has twofold contributions, which are both theoretic and empirical. The theoretic contribution is the unification of existing algorithms using Shapley-Taylor indices. The empirical contribution is their proposed fast-kernel algorithm, which has been validated through quantitative experiments.

Few concerns:

1. Readability. For the theoretic part, this work is built upon Shapley value. I would recommend authors to include more introduction for Shapley value (at least more intuition), so that readers do not need to read external references.

2. Please re-order table 1 that mIoU should be the primary metrics. It is because, without good mIoU, faithfulness and inefficiency is not that meaningful metrics.

3. Variance for Table 1?

4. Wondering do people really care about the speed of explanation?





**Summary Of The Paper:**

This paper proposes a unified framework to explain the current model explaining algorithms. Under this framework, the authors discovered the current algorithms are approximating second-order Shapley-Taylor indices, where they also proposed a fast-kernel-based estimation method. Empirically, the authors benchmarked different algorithms in two large segmentation datasets. They showed the proposed algorithms achieved better faithfulness and inefficiency.


**Summary Of The Review:**

This work has solid contributions, and the writing and experiments presentation could be improved.

---

> ### Author Response · Authors · 2021-11-19
> **Improved Readability, Clarifying Table 1, Additional Datasets, Explaining Dissimilarity**
>
> Thank you for your thoughtful comments on our work. We have updated our manuscript and marked major changes with yellow highlights.
>
>
> ## Readability:
> We appreciate your feedback on the Shapley Value introduction section and have added several sentences to provide some clarifying intuition. In particular, we have tried to better highlight the duality between paying employees in a company and explaining the predictions of a machine learning model. We have also given some intuitive descriptions of the axioms of fair credit assignment. We note that these simple and intuitive axioms imply that the Shapley value is the only fair credit assignment strategy you can have without giving up one of the properities (Symmetry, Linearity, Efficiency, and Dummy).
>
> ## Table 1 Clarifications:
> You mentioned that mIoU should be the primary metric but we wanted to clarify the relative importance of these metrics. We argue that it is Faithfulness that is the key metric as it is the only way to guarantee that the model explanation method is capturing the behavior of the underlying model. If the underlying model is not trained well, there is no reason to expect a model explanation strategy should yield results with a high mIoU (but we can still have a faithful explanation). Instead, we should aim to build model explanation methods that faithfully capture the dynamics of the model being explained, which is what the “Faithfulness” metric measures directly. We have included mIoU as a helpful indicator for those that wish to use model explanations for weakly supervised learning, but stress that this is not the only “proper” or “primary” way to evaluate a model explanation method.
> Regarding table variances, we omitted these variances because they are significantly smaller than the precision of the table and do not provide actionable information. We have added a section in the supplement that mentions this finding to be precise.
>
> ## On the Speed of Explanations:
> Computing Shapley values and Shaply-Taylor indices are generally NP-Hard. Thus, exact computation is prohibitively expensive for high-dimensional functions. So it is imperative that we develop faster approximations of these quantities to enable research on modern machine learning models. Furthermore, some authors have used model explanation methods to distill classification labels into localized predictions and heatmaps. These applications can require millions of function explanations and as a result, any accelerations to this method can dramatically speed downstream tasks.
>
>  ##  Additional Experiments on CU Birds and Stanford Online Products:
> Finally, we have added additional experiments on the CU Birds and Stanford Online Products datasets as requested. Though these datasets do not contain the necessary information to evaluate second-order explanations we have compared the performance of our first-order methods. In general, we find that explanations based on our fair credit assignments better capture the relevant areas of the image and yield a larger change in the model’s score when blurred. The methods also better satisfy the Efficiency axiom of fair credit assignment.
>
>  ## Explaining Dissimilarity:
> We have added Section A.15 in the appendix to highlight how our methods can explain image dissimilarity. In particular, areas with a negative Shapley value can be interpreted as lowering the overall similarity of the two images. Thus these negative values are the regions that contribute most to the “dissimilarity” of the two images.
>
> Thank you again for your thoughtful feedback and we kindly ask that you consider raising your score if you found these comments helpful.

---

> ### Author Response · Authors · 2021-11-29
> **Following Up**
>
> Thank you for your helpful comments, as today is the last day of the review please let us know if there is anything else we can address at this time. We appreciate your consideration

---

### Author Response · Authors · 2021-11-22
**Thanks**

Thank you to all the reviewers for the helpful comments and feedback. Please let us know if there is anything else that we can address prior to the review period closing. We appreciate your help to make this work better and will continue to work with you to ensure it meets your quality standards if accepted.

---

### Decision · Program_Chairs · 2022-01-20

**Decision:**

Accept (Poster)

**Comment:**

The initial reviews for this paper were 6,6,6, the authors have provided a rebuttal and after the rebuttal the recommendation stayed the same. The reviewers have reached the consensus that the paper is borderline but they have all recommended keeping it above the acceptance threshold. Following the recommendation of the reviewers, the meta reviewer recommends acceptance.